# A Novel Insight into the Ullmann Homocoupling Reactions Performed in Heterogeneous Catalytic Systems

**DOI:** 10.3390/molecules28041769

**Published:** 2023-02-13

**Authors:** Ágnes Mastalir, Árpád Molnár

**Affiliations:** Department of Organic Chemistry, University of Szeged, H-6720 Szeged, Dóm tér 8, 6720 Szeged, Hungary

**Keywords:** Ullmann reaction, homocoupling, C-arylation, copper, palladium, gold, nanopartcle, Pd complex, bimetallic catalyst

## Abstract

The Ullmann reaction has been reported to be the first cross-coupling reaction performed by using a transition metal catalyst. This reaction has been initially considered as the copper-catalyzed homocoupling of aryl halides, leading to the formation of symmetrical biaryl compounds via the generation of novel C–C bonds. Although this reaction has been extensively studied in recent decades and valuable results have been achieved, there are still considerable efforts focused on the development of novel catalytic systems, mild reaction conditions, and extended substrate scope. The mechanistic aspects of the Ullmann homocoupling reaction have also been investigated, as related to the introduction of new sustainable strategies and green procedures. The application of recyclable heterogeneous catalysts has been found to overcome most of the limitations associated with the harsh reaction conditions of the original Ullmann reaction. More recently, copper-based catalytic systems have also been replaced by palladium nanoparticles, ionic palladium species, gold nanoparticles, and palladium–gold bimetallic systems. In this review, current results reported on the Ullmann homocoupling reaction are discussed, with an emphasis on the development of novel catalytic systems, which can be efficiently used under heterogeneous conditions.

## 1. Introduction

Carbon–carbon (C–C) bond formations through cross-coupling reactions belong to the most versatile and useful procedures applied in synthetic organic chemistry. These reactions are typically performed by using transition metal catalysts [1,2,3,4,5]. As reported earlier by Ullmann, elemental copper has been the first transition metal used to catalyze C–C bond formation [6]. Copper-based catalysts are still frequently applied in such reactions, as they are cheap and readily available in most laboratories [7,8,9]. On the other hand, the application of Cu-based catalysts has been affected by several drawbacks including a limited scope of reactants, the poor solubility of copper salts in organic solvents, the low tolerance concerning other functional groups, the requirement of a stoichiometric amount of copper catalyst, and the harsh reaction conditions including high reaction temperatures and strong bases [10,11]. Accordingly, considerable efforts have been made to develop additional catalytic systems, which may be favorably employed under mild conditions [12,13,14,15,16]. The recent application of noble metal catalysts including palladium-based systems, gold nanoparticles, and bimetallic species has also been found to be efficient in the Ullmann coupling reactions [17,18,19].

Transition metal complexes applied as homogeneous catalysts are generally sensitive to moisture, oxygen, and high temperature. The instability of such complexes restricts their catalytic applications, as they often require inert atmospheres, deoxygenated solvents, and special reaction conditions. It has also been pointed out that these catalysts are difficult to remove from the reaction mixtures [20,21]. The application of heterogeneous metal catalysts offers a promising alternative for C–C coupling reactions, as these catalysts are less sensitive to the reaction conditions, can be readily separated from the reaction media and recycled. Nevertheless, the nature of the active sites of heterogeneous catalysts tends to be complicated to establish and it still remains a challenge in most cases [22].

The aim of the present paper is to review the recent progress on the application of novel catalytic systems developed for the Ullmann homocoupling reactions performed under heterogeneous conditions. Emphasis has been laid on the utilization of catalytically active copper, palladium, gold, and bimetallic particles.

## 2. Results

### 2.1. Copper-Based Catalysts

Before the discovery of Pd^0^-catalysed cross-coupling reactions, transformations including C–C bond formation, such as the Ullmann homocoupling reactions, have been predominantly carried out by using copper catalysts [23,24]. Despite the above-mentioned drawbacks, Cu-based catalysts have still been employed for the homocouplings of aryl halides, on both the laboratory and the industrial scale. Furthermore, asymmetric Ullmann homocoupling reactions may also be efficiently accomplished by using Cu catalysts [25,26,27]. The general mechanism of the Cu-catalyzed Ullmann homocoupling reaction is shown in Figure 1 [1].

In the first step, a copper (I) halide undergoes a reaction with a nucleophile in the presence of a base, which is required for the removal of the HX acid. This results in the formation of a copper nucleophile complex. The following step is the oxidative addition of an aryl halide, leading to the formation of the reaction intermediate, in which the oxidation state of copper has changed to Cu^III^. Reductive elimination of this complex provides the Ullmann homocoupling product and regenerates the initial CuX catalyst, which is able to participate in further catalytic cycles.

Sarkar et al. developed a new synthesis procedure for the preparation of Cu_2_O nanocrystals, which were subsequently employed in the Ullmann homocouplings of aryl halides [28]. The Cu_2_O nanostructures were prepared from an aqueous solution of copper(II) acetate containing glucose. After the addition of NaOH, the solution was aged for 10 min and then subjected to pulse mode sonication, which resulted in the formation of the product, characterized by XRD, TEM, EDX, FT-IR, and XPS measurements. It has been pointed out that the final hybrid structures of the Cu_2_O nanocrystals were strongly affected by the amount of added glucose. Structural investigation revealed that the Cu_2_O nucleation clusters formed during reduction rapidly grew to nano-scaled particles with an octahedral morphology. XPS studies indicated the presence of either Cu^I^ or Cu^0^, as related to the binding energy of 932.3 eV, and the presence of Cu^I^ was also confirmed by the X-ray-induced Auger spectroscopy spectrum (XAES). The Ullmann homocoupling reaction was carried out at 120 °C for 20 h, by using 2-(ethylamino)ethanol as a solvent. For the investigation of Cu_2_O catalysts with different morphologies, it was established that the highest activities were displayed by octagonal nanoparticles. For the homocoupling of iodobenzene, bromobenzene, and chlorobenzene, the conversions were 87.7%, 75.6%, and 61.6%, respectively, and the TOF values were 2216, 2035, and 1696 h^−1^.

The results of Sheldon’s hot filtration test [29] indicated that the removal of the catalyst from the reaction mixture afforded no further transformation. The hot filtrate was analyzed by ICP and only a negligible amount of Cu (in ppb level) was detected, which indicated that the sample was a heterogeneous catalyst.

In order to overcome the drawbacks of the classical copper-catalyzed Ullmann reactions, efforts have been made to develop novel synthesis procedures for efficient working catalysts. Xie et al. presented a novel method for the synthesis of ligand-free silica-supported Cu nanoparticles, to be employed in the Ullmann homocoupling reaction [30]. In the first synthesis step, colloidal silica nanospheres, prepared via a modified Stöber method [31], were dispersed in water. Next, an NH_4_OH solution was added dropwise to an aqueous solution of copper sulfate, producing a transparent cuprammonium solution. The addition of this solution to the silica dispersion resulted in the formation of copper-ammonia complex-decorated silica particles. Reduction of the copper precursor was performed by a NaBH_4_ solution, which afforded the product Cu-SiO_2_. The elemental analysis confirmed that the Cu content of the product was 2.56 wt% and TEM images indicated that the Cu^0^ nanoparticles were embedded inside the silica spheres. It should be noted that the particle size range (1.5–4.5 nm) and the mean particle diameter (2.5 nm) were typical of conventional supported metal catalysts and, therefore, it is reasonable to assume that the active species of Cu^0^-SiO_2_ were Cu^0^ nanoparticles, rather than Cu quantum dots, as suggested by the authors. The catalytic activity of Cu^0^-SiO_2_ has been demonstrated in the Ullmann homocoupling reaction of bromamine, performed in water at 90 °C, as shown in Figure 2.

The conversion of the reaction (58.25% and 96.37% at reaction times 0.5 h and 2 h, respectively, proved to be considerably higher than those obtained by applying a commercial Cu powder catalyst (10% conversion at 2 h). The turnover frequencies determined for Cu^0^-SiO_2_ and the Cu powder were 0.24 × 10^3^ h^−1^ and 0.04 × 10^3^ h^−1^, respectively. The efficiency of the Cu^0^-SiO_2_ catalyst was related to the uniform size distribution of the Cu^0^ nanoparticles. It has also been suggested that aggregation of the Cu^0^ nanoparticles was prevented via the confinement effect exerted by the regular framework of the silica support material. Furthermore, the porous structure and the hydrophilic character of SiO_2_ have been claimed to promote the diffusion of the aqueous systems into the pore channels of the support material. Recycling experiments gave evidence that the Cu^0^-SiO_2_ catalyst could be recovered and reused without an appreciable deactivation, which indicated good stability (96–92%, avg. conversion 95%, five runs). The Cu content of the catalyst did not diminish on repeated applications, as confirmed by ICP-AES measurements.

García et al. reported on the one-step synthesis of Cu^0^ nanoparticles on graphene films, followed by their selective oxidation to Cu^I^ species. The key issue of their synthesis method was claimed to be the oriented growth of the Cu^0^ nanoparticles [32]. Based on previous studies on the pyrolysis of chitosan at 1000 °C, leading to the formation of graphene films [33], the synthesis procedure was initiated by the pyrolysis of Cu^II^-chitosan films supported on quartz as a substrate. It was assumed that during pyrolysis, Cu^II^-chitosan films form Cu^0^ nanoparticles on graphene, as related to the spontaneous segregation of the two components. AFM images of the product indicated that Cu nanoplatelets with an average height of 3 nm were formed on the surface of the graphene film. In agreement with the results of these studies, field emission scanning electron microscopy (FESEM) images revealed the regular morphology and the uniform distribution of the copper nanoparticles, for which the size range was 5–20 nm and the mean particle diameter was 8 nm. XPS studies confirmed the presence of both Cu^I^ and Cu^II^ species on the surface of the graphene films, as related to binding energies 933 eV and 935 eV, respectively. Accordingly, the product was considered as a graphene-supported Cu_2_O species (Cu_2_O-G). The catalytic test reaction was the Ullmann homocoupling of iodobenzene, performed at 160 °C for 24 h, by applying Cu_2_O-G (180 μmol% Cu) as a catalyst, 2 mmol of KOCH_3_ as a base, and 1,4-dioxane as a solvent. Under such conditions, the conversion was 26.4% and the selectivity for the formation of the biphenyl product was 77.8%. It was established that during the reaction, Cu_2_O-G was detached from the quartz substrate, and the graphene film was exfoliated, therefore the catalyst could be recycled only once (for the second application, the conversion and the selectivity were 26.8% and 78.3%, respectively).

### 2.2. Pd-Based Catalysts

The direct homocoupling of aryl halides has been considered as the best method for the construction of aryl–aryl bonds. However, C–F bond activation has remained a challenge, as the C–F bond is considerably stronger than the C–Cl and C–Br bonds. As related to the remarkable thermal and chemical stability of organofluorine compounds, transition metal-catalyzed C–F bond transformations have been investigated less frequently [34]. Moghadamm et al. reported on the Ullmann homocoupling of unactivated aryl halides, including fluorobenzene, performed by using a novel magnetic heterogeneous palladium catalyst, synthesized by immobilization of palladium onto aminopyridine-functionalized Fe_3_O_4_ particles [35]. Magnetic Fe_3_O_4_ nanoparticles, obtained from FeCl_3_ and FeCl_2_ by co-precipitation, were used for the synthesis of Fe_3_O_4_-silica-coated nanoparticles, which were treated with 3-glycidyl(oxypropyl)trimethoxysilane, followed by the addition of 2-aminopyridine. The addition of K_2_PdCl_4_ to the resulting solid afforded the formation of the product denoted as Pd^0^-SiO_2_/Fe_3_O_4_. Structural characterization of the sample was performed by temperature-programmed reduction (TPR), transmission electron microscopy (TEM), and atomic absorption spectroscopy (AAS).

TEM images indicated that well-dispersed Pd^0^ particles were formed, and the average particle size was 10 nm. The sample was tested as a catalyst in the homocoupling reactions of aryl halides (Figure 3). It should be noted that the reaction of fluorobenzene was performed by applying a prolonged reaction time and no substituent effect was investigated for this reactant. Recycling of the catalyst was performed for the homocoupling of bromobenzene, and it was found that the catalyst could be recycled four times with a significant decrease in activity (92–79%, avg. yield 89.6%, five runs).

Somsook et al. synthesized metal-oxide-supported palladium-doped nanoparticles, which were subsequently utilized in the homocoupling reactions of aryl chlorides [36]. La_2_O_3_, employed as a support material, was prepared under basic conditions, via co-precipitation of ferrocenium with La^III^ ions. In order to obtain an activated structure of ferrocenated La_2_O_3_, calcination of the support was carried out, followed by impregnation with a solution of PdCl_2_, which resulted in the formation of Pd nanoparticles. The synthesis product, denoted as Pd^0^/Pd^II^-*ferro*-La_2_O_3_, was characterized by ICP-OES, TGA, BET, SEM-EDS, HRTEM, PXRD, and XPS measurements. The Pd content of Pd^0^/Pd^II^-*ferro*-La_2_O_3_ was found to be 1.0 wt% and the Pd particle size range was 1.7–3.1 nm, as revealed by EDX studies and HRTEM. XPS spectra indicated that both Pd^0^ and Pd^II^ species were formed on the surface of La_2_O_3_ (as revealed by binding energies ranging from 335.3 to 343.2 eV). The catalytic activity of Pd^0^/Pd^II^-*ferro*-La_2_O_3_ was investigated in the Ullmann homocoupling reactions of chloroarenes by using 2.5 mol% Pd as a catalyst, NaOH as a base, and ethanol as a solvent, which was suggested to work as a reducing agent. The results are indicated in Figure 4.

The catalyst was claimed to be recovered and reused up to five times without an appreciable loss of activity (98–94%, avg. yield 96.6%, five runs).

Li et al. disclosed a novel synthesis procedure based on the immobilization of PdCl_2_ on MOF-253 with 2,2′-bipyridine moieties [37]. MOF-253 was prepared in the solvent *N*,*N*′-dimethylformamide (DMF) at 120 °C for 24 h, via the hydrothermal reaction of 2,2′-bipyridine-5,5′-dicarboxylic acid with AlCl_3_·6H_2_O and acetic acid. The addition of PdCl_2_ to MOF-253 afforded the product, Pd-MOF-253, for which the Pd loading was 3.0 wt%. Structural characterization of the sample was affected by XPS, XRD, and BET measurements. The formation of Pd^II^ ions was confirmed by XPS spectra, which displayed a characteristic Pd 3d_5/2_ signal at 338 eV, and a shift of the N 1s peak. The latter was related to the coordination interaction between the Pd^II^ ions and the nitrogen atoms of the 2,2′-bipyridine groups. Pd^II^-MOF-253 was tested as a catalyst in the homocouplings of iodobenzenes and the results are summarized in Figure 5.

Hot filtration experiments indicated that Pd^II^-MOF-253 was a heterogeneous catalyst, as no further reaction was observed after the removal of the catalyst from the reaction mixture. It was also pointed out that the extent of Pd leaching was negligible (< 0.1%). The catalyst could be recycled, and no appreciable activity decrease was observed on repeated applications (97–94%, avg. yield 96%, five runs). It was also established by PXRD studies that the structure of MOF-253 underwent no significant changes under reaction conditions.

A ligand-free Pd-catalyzed Ullmann biaryl synthesis has been developed by Zhu et al., by using the combination of Pd(OAc)_2_ and hydrazine hydrate as a reducing agent at room temperature, applied for the homocoupling reactions of aryl iodides [38] (Figure 6).

It was found that an efficient reaction took place for both electron rich and electron-deficient aryl iodides, as well as for hetero-aryl iodides, leading to a wide range of biaryl products in good to excellent yields. It was suggested that the catalytically active species were Pd^0^ nanoparticles formed in the presence of hydrazine hydrate. The reaction was also performed for sterically hindered *ortho*-substituted aryl iodides (R = OH, CH_2_OH, NH_2_, NHCOCH_3_), for which the isolated yields were 68–83%. Furthermore, the reaction of 2-iodopyridine afforded a yield of 82%. The proposed mechanistic pathway is displayed in Figure 7.

The first reaction step is the reduction of the precursor Pd(OAc)_2_ to Pd^0^ by hydrazine hydrate, which provides catalytically active Pd^0^ nanoparticles. The following step is the oxidative addition of iodobenzene to the Pd^0^ particles, resulting in the formation of the complex ArPd^II^I, followed by transmetalation, which affords Ar-Pd^II^-Ar as an intermediate. Meanwhile, PdI_2_ is released and reacts with hydrazine hydrate to produce Pd^0^ nanoparticles. The final reaction step is reductive elimination, which results in the formation of the Ar-Ar coupling product and regenerates the active Pd^0^ species, which may participate in further reactions.

Barman et al. reported on the synthesis of highly dispersed Pd nanoparticles embedded in graphitic carbon nitride (CN) by an ultrasonic method, without using an external reducing agent [39]. Graphitic carbon nitride was obtained by the microwave heating of formamide at 180 °C and Pd nanoparticles were synthesized on the surface of this material via ultrasonication of an aqueous PdCl_2_ solution. The formation of Pd nanoparticles in the product, denoted as Pd^0^/Pd^II^-CN_x_, was confirmed by XRD studies. XPS spectra revealed the appearance of two asymmetric peaks at the binding energies 335.5 eV and 340.7 eV, assigned to the Pd^0^ state, whereas another pair of peaks at 336.6 eV and 341.8 eV indicated the formation of Pd^II^ species. It was also established that the major Pd species (62%) in Pd^0^/Pd^II^-CN_x_ was Pd^0^. TEM images indicated that the Pd nanoparticles were uniformly distributed on the surface of the carbon nitride sheets and the average particle diameter was 2 nm. The catalytic activity of the Pd^0^/Pd^II^-CN_x_ sample was investigated in the Ullmann homocoupling reactions of various aryl halides and the results are summarized in Figure 8.

It was found that haloarenes with both electron-donating and withdrawing groups provided the biaryl products with good to excellent yields, whereas a lower reactivity was experienced for chlorobenzene. The substrate scope was also extended to 2-bromonaphtalene and 2-bromopyridine, which afforded the corresponding coupling products with 72% and 54% yields, respectively, at extended reaction times of 24 h. Recycling of the catalyst was investigated for the reaction of iodotoluene and a considerable decrease in activity was observed (90–52%, avg. yield 73%, five runs).

For the production of recyclable heterogeneous catalysts, cyclodextrins have also been utilized as support materials for metal nanoparticles [40,41,42]. Bazgir et al. disclosed a novel synthesis procedure based on the formation of β-cyclodextrin-supported Pd nanoparticles, obtained under mild conditions [43]. The catalyst was prepared by adding an acetonitrile solution of Pd(OAc)_2_ to an aqueous solution of β-cyclodextrin at room temperature, which resulted in a chelate complex formation between the Pd^II^ ions and the secondary OH groups of cyclodextrin. Reduction of the precursor was performed by an aqueous solution of NaBH_4_, affording the product Pd^0^-β-CD, with a Pd loading of 40 wt%. EDS analysis of the sample confirmed the formation of Pd^0^ nanoparticles, also detected by TEM measurements, and XRD studies confirmed the face-centered cubic (fcc) structure of these particles. The average Pd particle diameter, 12–15 nm, proved to be larger than the cavity of β-cyclodextrin, which indicated that the particles were distributed on the external side of the cyclic cavities. The catalytic activity of Pd^0^-β-CD was investigated in the reductive Ullmann homocoupling reactions of aryl halides, by using 1 mol% of catalyst, K_2_CO_3_ as a base, and ascorbic acid as a reducing agent (Figure 9).

The catalyst was found to display a good performance for the transformations of both substituted aryl iodides and bromides. However, the homocoupling of aryl chlorides did not take place. Recycling of the catalyst was studied for the homocoupling of iodobenzene. It was established that the catalyst could be efficiently reused up to six cycles with only a moderate decrease in activity (90–83%, avg. yield 86.8%, six runs). The results of a hot filtration test indicated that the reaction took place via a heterogeneous pathway.

The application of magnetic nanoparticles has been found to facilitate the recovery of heterogeneous catalysts. Fe_3_O_4_ nanoparticles incorporated into various support materials including silica, dopamine, carbon, and polymers, have been reported to improve the catalytic activity of metal nanoparticles by preventing their aggregation [44,45,46]. In a recent study, Dubey and Kumar reported on the preparation of polydopamine-coated iron oxide nanoparticles, applied as the support material of Pd catalysts in the Ullmann homocoupling reaction [47]. The synthesis procedure was based on the self-polymerization of dopamine, resulting in the in situ formation of polydopamine-coated Fe_3_O_4_ particles, applied as a support material of catalytically active Pd nanoparticles. Structural characterization of the product, denoted as Pd^0^-PDA-Fe_3_O_4_, was performed by various instrumental techniques including XPS, XRD, TEM, SEM, and ICP-AES. The Pd content of the product was 6.81 wt%, as confirmed by EDX analysis, and TEM images indicated the appearance of Pd nanoparticles, for which the size range was 2–5 nm. XPS spectra displayed Pd 3d_5/2_ and 3d_3/2_ signals at binding energies 335.31 eV and 340.51 eV, which referred to the formation of Pd^0^ species. The Ullmann homocoupling reactions of aryl halides were performed at 100 °C for 24 h, in the presence of glucose, by applying Cs_2_CO_3_ as a base and randomly methylated β-cyclodextrin (RM-β-CD) as a mass transfer reagent. The results are displayed in Figure 10.

The catalyst was also tested for the cross-coupling reactions of two different aryl halides. For the reaction of 4-iodoanisole and 4-iodotoluene (1:2), performed under the same conditions as those in Figure 10, the isolated yield of the cross-coupled product was 72%. After the reactions have been completed, the catalyst was recovered with an external magnet and recycled. It was found that the catalyst could be reused up to five times with only a minor decrease in activity (92–83%, avg. yield 87%, five runs). Pd leaching proved to be negligible, as the Pd concentration of the reactant mixture was 1.4 and 7.2 ppm after the first and fifth cycles, respectively. A hot filtration test was also performed, and the filtered solution displayed no further activity, which confirmed that the Pd^0^-Fe_3_O_4_/PDA sample was a heterogeneous catalyst.

Figure 11 displays the reaction mechanism proposed for the Ullmann homocouplings of aryl halides. The first step involves a reactant cyclodextrin inclusion complex interaction with the Pd nanoparticles, followed by the oxidative addition of the aryl halide on Pd^0^, which produces an arylpalladium halide intermediate. At this point, the reaction may proceed through two pathways. The first path involves the addition of another molecule of aryl halide to the intermediate, which results in the formation of a diarylpalladium complex and the release of the halide ions. This is followed by the elimination of the coupling product, facilitated by the glucose molecules, which regenerate the active Pd^0^ species. An alternative pathway is the reaction of an aryl halide with the intermediate, affording the Ullmann product, together with PdX_2_. Reduction of PdX_2_ with glucose results in the formation of Pd^0^, and hence the catalytic cycle is completed. Meanwhile, glucose is oxidized to gluconic acid [48].

A novel synthesis protocol for the preparation of Pd nanoparticles on N-doped graphene nanocomposites has been described by Bazgir et al. [49]. Graphene oxide was first synthesized via the oxidation of graphite powder by the Hummers method [50]. N-doped graphene sheets (NG1) with an enhanced N content of 11.24 wt% were obtained by treating an aqueous GO dispersion with urea at 180 °C for 12 h. The solid was subsequently dispersed in a mixture of ethylene glycol and water, followed by the addition of PdCl_2_ at pH = 11. Reflux of the mixture at 110 °C for 7 h was suggested to result in a reduction of the precursor PdCl_2_. For comparison, another NG material with a lower N content of 3.2 wt% (NG2) was also prepared and loaded with Pd in a similar procedure as described above. The Pd loadings of the products Pd-NG1 and Pd-NG2 were 0.80 and 0.74 mmolg^−1^, corresponding to 8.5 and 7.9 wt% Pd, respectively. Raman spectra of the samples indicated that nitrogen doping of graphene tends to enhance the Pd binding to the defect sites of the support, thereby lowering the occurrence of metal–metal interactions [51]. XPS analysis of Pd-NG1 confirmed the formation of Pd^0^ species, as related to the binding energies 335.48 eV and 340.83 eV, corresponding to the Pd^0^ 3d_5/2_ and 3d_3/2_ peaks, respectively. TEM micrographs revealed the formation of highly dispersed Pd nanoparticles on NG1, with an average particle diameter of 15 nm. The catalytic performance of Pd^0^-NG1 was studied in the Ullmann homocoupling reactions of various aryl halides without using an external reducing agent [52]. It was suggested that N-doped graphene, as a polycyclic aromatic molecule, acted as both a support material and a reducing agent in the reaction. The results are displayed in Figure 12.

The Pd-NG1 sample proved to be an efficient catalyst for the homocoupling reactions of aryl iodides. For the transformations of aryl bromides and chlorides, good to excellent biaryl yields were obtained at prolonged reaction times. Recycling of the catalyst was investigated for the homocoupling of iodobenzene, and it was found that the catalyst could be reused up to five times without a significant loss of activity (98–84%, avg. yield 92%, five runs).

As reported earlier by Karimi et al., ionic liquid-derived nanofibrillated mesoporous carbon (IFMC) can be efficiently used as a support material for Pd nanoclusters [53]. In a recent study, it has also been demonstrated that the synergistic effect between the nitrogen-containing functional groups of the ionic liquid and the supported Pd nanoclusters improved the catalytic performance of this sample in the Ullmann homocoupling reactions of aryl halides [52]. IFMC has been obtained by carbonization of 1-methyl-3-phenethyl-1H-imidazolium hydrogen sulfate (MPIHS), by applying SBA-15 as a hard template [53]. The supported Pd catalyst, Pd^0^-*nanofibr*-mesoC, was subsequently obtained by the impregnation of IMFC with a THF solution of Pd(OAc)_2_, followed by reduction with a solution of hydrazine hydrate. The product was characterized by N_2_ sorption, XPS, and TEM measurements. It was found that the Pd nanoparticles were supported not only on the external surface of the sample but also inside the mesopores. The mean particle diameter was 4–5 nm. XPS studies revealed a doublet peak at binding energies 335.2 and 340.3 eV, associated with Pd 3d_5/2_ and Pd 3d_3/2_ signals, respectively, which clearly indicated the formation of Pd^0^ species. It was also established from the N1s signals that both pyrrolic and pyridinic nitrogen were present in the product. The catalytic performance of Pd^0^-*nanofibr*-mesoC was investigated for the homocouplings of iodobenzenes in the absence of a reducing agent (Figure 13).

For the reactions of substituted bromoarenes and chloroarenes, bearing either electron-donating or electron withdrawing groups, excellent yields were obtained for the respective biaryl products and for the latter reactants, no hydrodechlorination by-products were detected under the experimental conditions. The catalyst also displayed a considerable activity for the transformations of heteroaryl bromides. For the homocouplings of 2-bromotiophene and 3-bromopyridine, the product yields were 58% and 78%, respectively. These reactions have been claimed to be the first examples for the Ullmann homocouplings of heteroaryl halides in water, in the absence of a co-reductant. The homocoupling of 4-iodoanisole was also tested in a five-run reuse study applied under the reaction conditions shown in Figure 13. Continuous and significant decrease in the activities was experienced (99–85%, avg. yield 94%). It is obvious that this phenomenon is related to a large (about an 80%) drop of the Pd content.

Ohtaka et al. synthesized Pd nanoparticles supported on linear polystyrene, to be applied as a catalyst in the Ullmann homocoupling reactions of aryl halides [54]. The catalyst was prepared under basic conditions, from a mixture of polystyrene, Pd(OAc)_2_, and 4-methylphenylboronic acid, which was subjected to magnetic stirring at 90 °C for 5 h. XPS spectra of the product, Pd^0^-PS, displayed characteristic peaks at binding energies 335.0 and 340.2 eV, assigned for Pd 3d_5/2_ and Pd 3d_3/2_, respectively, indicating the formation of Pd^0^ species. TEM images revealed that the Pd nanoparticles were dispersed on the surface of polystyrene and the mean particle diameter was 2.7 nm. The Ullmann homocoupling reactions were performed in the presence of methanol, which was applied as a reducing agent.

It was found that both electron-rich and electron-deficient aryl bromides afforded the coupling products with good yields, whereas the catalyst proved to be less efficient for the transformations of chlorobenzene and iodobenzene (Figure 14).

Recycling of the catalyst was investigated for the homocoupling of 4-bromotoluene, and a considerable deactivation was observed, which was attributed to Pd leaching (85–37%, avg. yield 62.5%, four runs).

As related to their favorable properties, hybrid organic–inorganic polymers have recently attracted considerable attention. However, there have been only a few reports on the catalytic applications of these materials [55,56]. Mossadegh and Yavari described a synthesis procedure for the preparation of a novel polymer–inorganic hybrid, which was employed as a support material for Pd nanoparticles [57]. In the first synthesis step, a mixture of MCM-48 and N-vinyl-2-pyrrolidone (NVP) was treated with benzoyl peroxide, followed by stirring under reflux conditions, which afforded poly(N-vinyl-2-pyrrolidone)-MCM48 (PVP-MCM-48). Then, the aqueous solution of the precursor Pd(OAc)_2_ was added to this solid, and the mixture was subjected to magnetic stirring at 80 °C for 5 h. Reduction of the precursor was performed by hydrazine hydrate. The Pd content of the product, Pd^0^-PVP/MCM-48, determined by ICP-AES, was 12.1 wt% (1.14 mmolg^−1^). XRD studies revealed that polymerization and Pd immobilization had no significant effect on the structure of MCM-48. The results of FTIR measurements indicated that the Pd nanoparticles were coordinated to the support through the C=O groups of polyvinylpyrrolidone. The absence of the characteristic UV-Vis peak of Pd^II^, determined for Pd(OAc)_2_, confirmed that complete reduction of the precursor took place. This was also confirmed by XPS spectra, which displayed characteristic peaks at binding energies 333.8 eV and 339.1 eV, assigned to the Pd 3d_5/2_ and Pd 3d_3/2_ core levels of the Pd^0^ species. The Pd^0^-PVP/MCM-48 sample was investigated as a catalyst in the Ullmann homocoupling reactions of aryl halides under optimized reaction conditions and the results are shown in Figure 15.

The catalyst exhibited good performance for the transformations of aryl iodides and bromides, whereas a lower activity was observed for the reaction of aryl chloride. Recycling experiments, carried out for the homocoupling of iodobenzene, indicated a high catalyst stability with only a minor Pd leaching of 3.5% (90–85%, avg. yield 88%, seven runs). This was attributed to the stabilizing effect of both MCM-48 and PVP, which were suggested to prevent particle aggregation.

As reported earlier, the combination of C-C coupling reactions with continuous flow conditions had several drawbacks, including metal leaching, which was attributed to the gradual removal of the metal by the solvent during time-on-stream [58]. Therefore, attempts have been made for the development of catalysts with enhanced stabilities, which may be efficiently employed in C–C coupling reactions performed in continuous flow reactors under mild conditions. An environmentally friendly procedure for the preparation of biaryls under continuous flow conditions has been reported by Luque et al. [59]. The reductive homocoupling reactions of aryl halides were carried out by using commercially supported Pd catalysts, of which 5% Pd^0^-C was applied in most cases. It was found that the homocouplings of both iodobenzene and bromobenzene (0.1 mol) could be efficiently conducted at room temperature at a low hydrogen pressure of 2 bar, after 42 s of residence time and 20 min on stream, by applying 5% Pd^0^-C as a catalyst, 0.25 mol K_2_CO_3_ as a base and the solvent mixture MeOH:H_2_O = 3:1. Under such conditions, the conversions obtained for both reactants exceeded 99%. Recycling studies effected for the transformation of iodobenzene gave evidence that the extent of Pd leaching was negligible (5–10 ppm) and the catalyst could be reused with only a minor loss of activity (99–98%, avg. yield 98.6%, 10 runs).

Homocoupling reactions have been generally considered insensitive to light and therefore the application of photocatalysts for these reactions has not yet been extensively studied. Nevertheless, the combination of the electron transfer of photocatalysts with heterogeneous catalysis has recently gained in importance. It has been suggested that reagents with lower reactivities perform better under visible light irradiation [60,61,62]. In a recent study, Rezaeifard and coworkers explored the utilization of a catalyst with Pd^0^ particles deposited onto TiO_2_ nanoparticles [63]. The catalyst with a 2.12 wt% Pd loading was fabricated by treating TiO_2_ with ascorbic acid, followed by a reaction with Pd(OAc)_2_ in ethanol. Note that both steps were carried out under sonication. Ascorbic acid has a double role. It lowers the band gap energies, thereby promoting the absorption of visible light, and it stabilizes Pd particles generated by sonication. The results of appropriate characterization methods (FT-IR, XPS, TEM) confirmed successful catalyst synthesis. This sample was applied as a photocatalyst for the Ullmann homocoupling reaction under visible light irradiation and solvent free conditions. The addition of an aqueous solution of ascorbic acid (AA) to TiO_2_ nanoparticles, obtained by a sol–gel procedure, followed by stirring at rt for 8 h, afforded an AA-TiO_2_ nanohybrid material. This was added to the precursor Pd(OAc)_2_ dissolved in ethanol under ultrasonic agitation, and then the mixture was refluxed for 12 h. Structural characterization of the Pd^0^-AA-TiO_2_ material was performed by various instrumental methods. The Pd content of the sample was 2.12 wt%, as determined by ICP-AES. FTIR spectra indicated the formation of Pd-O bonds, assigned to the coordination of Pd to the AA-coated TiO_2_ particles. XPS measurements gave evidence that the binding energies of Pd 3d_3/2_ and Pd 3d_5/2_ were 340.54 eV and 335.3 eV, respectively, which confirmed the formation of Pd^0^ particles.

The catalytic performance of the Pd^0^-AA-TiO_2_ nanohybrid material was tested in the Ullmann homocoupling reactions of aryl halides. By applying a low Pd loading of 0.3 mol%, the corresponding biaryls were produced with 65–95% yields. On the other hand, the reactions of aryl bromides and aryl chloride required prolonged reaction times (Figure 16). The results of recycling experiments, obtained for the homocoupling of iodobenzene, indicated that the catalytic system could be reused six times without a significant loss of activity (95–90%, avg. yield 92.5%). A hot filtration test also indicated that Pd leaching did not take place and hence the Pd^0^-AA-TiO_2_ nanohybrid was a heterogeneous catalyst.

The immobilization of homogeneous catalysts on solid support materials by different methods including encapsulation, coordination, and ion interaction, has been found to exert a beneficial effect on the recovery and reuse of these catalysts [64,65,66]. Covalent attachment has also been reported to be a particularly efficient method for the immobilization of metal complexes with phosphine and bisphosphine ligands on support materials containing surface OH groups [67]. The grafting of an aminobis(phosphine)–Pd^II^ complex [PdCl_2_{(Ph_2_P)_2_N(CH_2_)_3_Si(OMe)_3_}] on graphene oxide (GO) as a support material has been carried out by Balakrishna et al. [68]. Immobilization of the Pd complex on GO was performed in toluene at 110 °C for 24 h under a N_2_ atmosphere. The synthesis procedure was based on a condensation reaction between the methoxysilane groups of the Pd complex and the hydroxy groups of GO. The resulting composite material, denoted as Pd^II^-PNP-GO was characterized by FTIR spectroscopy, solid-state ^31^P NMR, SEM, TEM, XPS, and ICP-AES techniques, which confirmed that the Pd complex was successfully immobilized on the surface of GO. XPS studies also gave evidence that the oxidation state of Pd in the composite material was +2 (the binding energies for Pd 3d_3/2_ and Pd 3d_5/2_ signals were 341.5 eV and 336.2 eV, respectively). The catalytic performance of Pd^II^-PNP-GO with a Pd loading of 2.8 wt% was investigated in the Ullmann homocoupling reactions of substituted aryl chlorides and a substantial catalytic activity was experienced, as the corresponding biaryl compounds were produced with high yields. The results are shown in Figure 17, which also displays the structure of the catalyst.

Recycling studies, effected for the homocoupling of 3-chlorotoluene, indicated moderate stability of the catalyst, as related to a gradual decrease in activity observed on repeated applications (94–76%, avg. yield 85%, five cycles).

The incorporation of Pd nanoparticles in *N*-heterocyclic carbene (NHC)-based organometallic polymers has been found to result in the formation of efficient catalysts, applied in C–C coupling reactions [69,70]. Recently, Karimi et al. disclosed additional results on the catalytic application of a main-chain NHC-based Pd catalyst in the Ullmann homocoupling reaction [71]. In a three-step catalyst synthesis, first 1,2,4,5-tetraaminobenzene and formic acid were reacted to form benzobis(imidazole) (110 °C, 36 h). In the next step, alkylation delivered bis(imidazolium) bromide (NaH, toluene, 110 °C, then 1-bromododecane, 25–110 °C 4 h and 110 °C 6 h). The final catalyst was prepared by using the precursor Pd(OAc)_2_ (DMSO, 110 °C, 10 h).

The Pd^II^-NHC polymer, for which the Pd loading was not reported, was applied as a catalyst in the Ullmann homocoupling reactions, for a limited scope of aryl halides, as shown in Figure 18. It was found that the corresponding biaryls were produced with good to excellent yields. Nevertheless, it should be noted that the transformations of chlorobenzenes afforded lower product yields, despite the application of an elevated temperature, prolonged reaction time, and KI as an additive. The scope of the reactants was subsequently extended to heteroaryl compounds, including 3-bromopyridine, 3-iodopyridine, 3-bromotiophene, and 2-bromotiophene. Under the mild reaction conditions indicated in Figure 18, excellent product yields of 82–95% were obtained. Recycling experiments were performed for the homocoupling of bromobenzene, and it was established that the catalyst could be reused without an appreciable decrease in activity (92–77%, avg. yield 86%, five runs).

### 2.3. Gold Catalysts

As related to its considerable polarity, wide solubility range, and reducing ability, *N*,*N*-dimethylformamide (DMF) has been found to exert a beneficial effect on C–C coupling reactions catalyzed by metal nanoparticles [72]. In order to extend the catalytic applications of DMF-protected gold nanoparticles [73], Lang et al. synthesized DMF-stabilized Au nanoparticles, to be utilized as catalysts in the Ullmann homocoupling reactions [74]. The Au nanoparticles were prepared via the addition of an aqueous solution of HAuCl_4_ to preheated DMF, followed by reflux at 150 °C for 2 h. The resulting yellow solution was found to contain Au nanoparticles, for which the mean particle diameter was 2.5 nm, as revealed by HRTEM measurements. PXRD studies also confirmed the formation of monodispersed Au^0^ nanoparticles, for which the lattice fringes were consistent with those of metallic gold and corresponded to the lattice spacing of the (111) crystal plane of face-centered cubic Au^0^ particles.

XPS spectra also revealed the appearance of Au^0^, indicated by 4f_7/2_ signals at 84.7 eV and 88.3 eV. In this study, the homocouplings of iodobenzenes were carried out delivering substituted biphenyls in prolonged treatments (48 h), providing moderate to excellent product yields in most cases (Figure 19). However, both the electronic effects and the steric hindrances of the substituents affected the productivity of the reactions. Compare, for example, the yields of 1-chloro-4-iodobenzene (95%) and 1-iodo-4-methoxybenzene (85%). The yield of sterically hindered 1-iodo-2-methoxybenzene was 43%, whereas the lowest reactivity with a yield of a mere 15% was obtained for 1-iodo-3,5-dimethylbenzene. The reaction of 2-iodopyridine afforded the coupling product with a yield of 50%, whereas the catalyst proved to be inefficient for the homocoupling of bromobenzene. The reusability of the catalyst was examined for the homocoupling of iodobenzene, and a considerable deactivation was observed (91–43%, avg. yield 71%, five runs).

So far, there have been rather few reports on the Ullmann homocoupling reactions catalyzed by gold nanoparticles [75,76]. The preparation of highly dispersed gold nanoparticles, supported on a magnetic mesoporous silica material, has been reported by Rostamizadeh et al. [77]. The synthesis procedure of the catalyst, reported in a previous paper, included the preparation of (α-Fe_2_O_3_)-MCM41 from a mixture of a magnetic Fe_3_O_4_ colloid, sodium silicate, and cetyltrimethylammonium bromide (CTABr), followed by calcination. This material was functionalized by 3-mercaptopropyl-trimethoxysilane, followed by the addition of the precursor HAuCl_4_, applied in a methanol solution. The Au loading of the product, Au^0^-α-Fe_2_O_3_/MCM-41, was reported to be 6.44 wt%. Structural characterization of the sample, performed by N_2_ sorption, SEM, and TEM measurements, indicated the formation of ordered mesopores and a uniform size distribution of the gold nanoparticles of 2–3 nm. Further, XRD patterns revealed the formation of Au^0^ nanoparticles with an fcc structure [78]. The sample was tested as a catalyst in the Ullmann homocoupling reactions of aryl iodides. The reactions were performed at 140 °C for 6 h, by using DMF as a solvent and K_3_PO_4_ as a base, as displayed in Figure 20.

For the reactions of substituted aryl iodides, the biaryl products were obtained with good to excellent yields. On the other hand, the transformation of aryl bromide did not take place, even at an elevated reaction temperature of 150 °C and a prolonged reaction time of 24 h. After reaction, the catalyst could be readily separated and recovered by an external magnet. Recycling of the magnetic nanocatalyst was investigated for the homocoupling reaction of 4-iodoacetophenone. It was found that the catalyst displayed high stability and could be reused without a significant decrease in activity (90–80%, avg. yield 84.7%, six runs). On the other hand, the reaction mechanism proposed by the authors cannot be fully justified.

### 2.4. Bimetallic Catalysts

A mixed metal oxide obtained from a Cu-Fe layered double hydroxide (LDH) via thermal decomposition has been reported to be an efficient ligand-free catalyst for the synthesis of biaryls through the Ullmann homocoupling reaction [79]. The Cu-Fe LDH precursor was synthesized by co-precipitation, from the precursors copper nitrate and ferric citrate under basic conditions [80]. Calcination of the resulting solid, performed at 600 °C for 4 h, resulted in the formation of a Cu-Fe mixed metal oxide (Cu-Fe-MMO), which was characterized by various instrumental methods including N_2_ sorption, TGA, XRD, XPS, and SEM/EDX measurements. XRD studies indicated the formation of a novel crystalline phase, corresponding to the Cu-Fe mixed oxide and TGA analysis confirmed that a phase transformation of LDH to a mixed metal oxide took place at 400 °C. XPS patterns revealed that the binding energy of the Cu 2p_3/2_ level was 934.4. eV, indicating the formation of a CuO phase, whereas the binding energy of the Fe 2p_3/2_ core level appeared at 712 eV, which was related to the presence of iron oxide. It was therefore suggested that CuO-Fe^III^-MMO consisted of a crystalline CuO phase, doped with Fe^III^ ions, derived from the LDH precursor. The results obtained for the Ullmann homocoupling reactions of aryl halides, performed for a limited scope of reactants, are shown in Figure 21.

It was found that the transformations of aryl iodides and bromides afforded good yields for the biphenyl products, whereas the reactions of chlorides and fluorides did not take place, even at prolonged reaction times. Recycling studies, effected for the reaction of 4-iodoanisole, indicated that the catalyst could be reused without an appreciable decrease in activity (80–78%, avg. yield 79%, five runs).

Chitosan (CS), considered the second most important natural polysaccharide after cellulose, has been used as a support material of transition metal catalysts, as related to its significant amount of NH_2_ and OH groups, which readily form chelate complexes [81]. In a recent study, Zeng et al. combined the advantages of chitosan and silica as support materials for the development of a heterogeneous membrane catalyst containing Pd and Zn nanoparticles [82]. A CS/SiO_2_ mixture was first prepared from an acetic acid solution of chitosan and a silica colloid, followed by the addition of Zn powder and a Na_2_PdCl_4_ solution under stirring. The mixture was subsequently dried to form membranes, which were purified by soaking in a NaOH solution and dried at 60 °C. The product was characterized by N_2_ sorption, FTIR, XRD, XPS, SEM, and HRTEM measurements. Structural investigation revealed that the surface of the membrane had an open macroporous structure, whereas the cross-section of the membrane had fewer pores with smaller sizes. XRD patterns displayed characteristic diffraction peaks of Zn^0^ and Pd^0^ particles and XPS spectra revealed two significant peaks at binding energies 337.5 and 335 eV, corresponding to Pd 3d_5/2_ signals, which indicated that the Pd species were both in zero-valent and divalent states. The catalytic activity of the Pd^0^/Pd^II^-Zn^0^-CS/SiO_2_ sample was investigated in the Ullmann homocoupling reactions of aryl iodides, performed in a DMSO-ethylene glycol solvent mixture at 120 °C, by using 5 mol% of catalyst. The results are indicated in Figure 22.

As shown in Figure 22, the Pd^0^/Pd^II^-Zn^0^-CS/SiO_2_ membrane catalyst exhibited a high catalytic activity. This finding may be associated with the incorporation of the active species into SiO_2_. Specifically, the generation of the porous structure of the CS/SiO_2_ matrix resulted in improved diffusion properties and high thermal stability. In addition, the active metal species were well distributed in the support. Catalyst reuse was explored in the homocoupling of iodobenzene in a five-run test with easy recovery. Results show a 10% drop-in activities, which is rather significant (97.8–87%, avg. yield 92.4%). This phenomenon is certainly attributed to the loss of Zn content (7.1%), resulting in the deterioration of the CS/SiO_2_ matrix. Note also the significant Pd leaching of 8.6 wt%.

Palladium-catalyzed Ullmann reactions have been used not only for the coupling reactions of simple organic molecules but also for the synthesis of complex organic molecules such as pharmaceuticals, polymers, and alkaloids [83,84]. As demonstrated by the studies included in the present contribution, palladium has been used as an efficient catalyst in most cases. Note, however, that palladium catalysts, utilized in homogeneous systems, almost always required the application of suitable ligands. Based on this information, Li, Zhang, and others suggested the application of a second metal or a suitable support material with the aim to improve the activity and the stability of the catalyst [85]. Therefore, here, we discuss two examples of the utilization of Au–Pd bimetallic catalysts, to be employed in Ullmann homocoupling reactions.

Ullmann homocouplings were affected by Zhang et al. with the application of a gold–palladium single-atom catalyst, specifically, by using an Au^0^_6_Pd^0^-resin [86]. In the catalyst synthesis, an anion exchange resin 717 (a cross-linked copolymer of benzene and styrene with a quaternary amine) was added to the aqueous solutions of HAuCl_4_ and H_2_PdCl_4_, followed by stirring for 2 h. The final catalyst was obtained by reduction with NaBH_4_. This synthesis process was performed with different Au–Pd ratios (AuPd, AuPd_2_, Au_x_Pd, x = 2, 4, 6, 8, 10, 20) with a total metal loading of 2 wt%. Various instrumental methods (XRD, ICP-AES, HRTEM, DTRIFTS) were used to collect useful pieces of information on the catalyst structure. According to EXAFS and CO adsorption measurements, bimetallic alloys were formed (particle size of 2–4 nm) with palladium islands separated by gold atoms. DRIFTS and EXAFS studies indicated that the Au–Pd single-atom structure is generated in the case of Au/Pd ≥ 4. Accordingly, the best yields and selectivities in chlorobenzene homocoupling were obtained with Au_6_Pd and Au_20_Pd, for which the TON values were 643 and 1148, respectively. The catalytic performance of the Au^0^_6_-Pd^0^-resin single-atom catalyst was investigated in the Ullmann homocoupling reactions of various aryl halides and high to excellent yields were obtained (Figure 23). For the samples with various Au–Pd ratios, a strong synergistic effect was established between the two metals. Namely, the highest activity was observed for Au^0^_6_-Pd^0^, whereas for the catalysts with enhanced Au–Pd ratios, a dehalogenation side reaction was found to decrease the productivities.

It was suggested that the active sites of the reaction were isolated Pd^0^ single atoms surrounded by Au^0^ atoms. A hot filtration test revealed that a heterogeneous reaction occurred, and recycling studies indicated that no significant activity decrease took place on repeated applications (97–93%, avg. yield 96.5%, eight runs).

Karimi et al. fabricated nanofibrillated mesoporous carbon support derived from the ionic liquid 1-methyl-3-phenethyl-1*H*-imidazolium hydrogen sulfate (MPIHS) [87]. In a carbonization process, MPIHS mixed with guanine (10% or 15%) was treated under argon in the presence of SBA-15 as a template at 900 °C. After removal of the template, the resulting nitrogen-rich materials (NMCI-1 9.2 wt%, NMCI-2 12.6 wt%), were loaded with gold and palladium by the reduction of Na_2_AuCl_4_ and Na_2_PdCl_4_ with NaBH_4_. N_2_ sorption isotherms of the samples indicated the formation of uniform mesoporous structures. The synthesis of the mesoporous carbon-stabilized bimetallic nanoparticles (Au^0^-Pd^0^-NMCI-2) was performed by the addition of NaPdCl_4_ and Na_2_AuCl_4_ to an aqueous suspension of the support material, followed by reduction with NaBH_4_. The total loadings of Au and Pd in the catalyst were 0.085 and 0.238 mmolg^−1^, corresponding to 0.157 wt% Au and 2.53 wt% Pd, respectively. XPS patterns displayed binding energies 335 and 340 eV, corresponding to Pd^0^, as well as 84 and 88 eV, corresponding to Au^0^, which confirmed that both metals were in zero oxidation states.

TEM analysis revealed that the metal particles were uniformly distributed over the surface of the support material. It was also established that the ratio of Au:Pd was 4:1 and the absence of any surface plasmon absorption peak related to the pure Au nanoparticles in the DRIFT-UV spectrum of Au^0^-Pd^0^-NMCI-2 clearly showed that the nanoparticles consisted of an Au–Pd alloy and not a mixture of Au and Pd species. The catalytic performance of the sample was investigated in the Ullmann homocoupling reactions of aryl chlorides and bromides (Figure 24), and significant activities were experienced. Recycling studies indicated that the catalyst remained efficient in repeated applications (95–88%, avg. yield 92.6%, five runs). However, after the fifth cycle, the total metal content was reduced to 85%, indicating that under reaction conditions, soluble catalytic species were generated, which could have reacted through a homogeneous pathway.

## 3. Conclusions

The Ullmann homocoupling reaction has been found to play an important role in organic syntheses and medicinal chemistry and can be widely employed for the preparation of biologically active compounds and pharmaceuticals. Although copper-mediated Ullmann homocouplings have remained the subject of extensive studies, the application of Pd-based systems, including Pd^0^ nanoparticles and Pd^II^ complexes immobilized on solid support materials, has clearly gained increasing importance. Heterogeneous Pd catalysts offer versatile applications, and their beneficial effect has been confirmed by their enhanced performances, as compared with those of copper-based catalysts. Furthermore, Pd-based catalytic systems can be employed under milder reaction conditions and for a broader substrate scope. In this respect, promising results have also been achieved for the utilization of gold nanoparticles and bimetallic systems, for which future prospects can be anticipated. Successful efforts have been made on the development of ligand-free catalysts and mild reaction conditions, including lower temperatures and/or green solvents, which give rise to the introduction of environmentally friendly procedures.

Despite the substantial progress reported for the investigation of the Ullmann homocoupling reaction so far, limitations in terms of substrate scope, catalyst efficiency, stability, and reuse still exist. For the transformations of aryl chlorides and fluorides, increased reaction temperatures, high metal loadings, and extended reaction times are required. Therefore, it may be suggested to introduce more efficient catalytic systems, which would also be suitable for practical applications. As the recent development of copper-catalyzed homocoupling reactions is far behind those promoted by Pd-based systems, it is also desirable to develop more sophisticated and more reactive copper catalysts, which may also be employed under mild conditions. Moreover, copper-catalyzed asymmetric coupling reactions still remain a challenge, which should be addressed by further studies.

In summary, a future perspective would be the development of improved catalytic systems with novel ligands, anchoring species, and/or support materials, which could be more efficiently used under milder conditions for an extended substrate scope, including less active chlorobenzene and fluorobenzene derivatives. The results reported for catalyst recycling also need considerable improvement and therefore the design of novel catalysts with enhanced activities and stabilities would also be appreciated.

## Data Availability

Data reported in this review are available in the papers listed in the References.

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
