# Peer review of "A Novel Insight into the Ullmann Homocoupling Reactions Performed in Heterogeneous Catalytic Systems"

_molecules, 2023, doi:10.3390/molecules28041769_

Round 1

Reviewer 1 Report

In this review authors have summarized the recent advancements made in the Ullmann homocoupling reactions over different heterogeneous catalysts. Results have merit. Thus, I am recommending this manuscript for publication after a revision considering the comments given below:

1) Authors should discuss more on mechanistic aspects of the homocoupling reactions. Different oxidation states of the metals and their characterizations (XPS) should be discussed.

2) One general mechanism on Cu-catalyzed Ullmann homocoupling should be provided.

3) Scheme 10: In the figure caption authors should mention that this homocoupling reactions are carried out over Pd-catalysts.

Author Response

We would like to thank the Reviewer for the positive evaluation of our manuscript and also for his helpful comments and suggestions.

Our answers to the comments are as follows.

  1. This review has been constructed from individual papers, in which different instrumental methods have been used for characterization. Most of the papers discussed in our manuscript reported XPS spectra and their evaluation. We agree with the Reviewer that the oxidation state of the metal is very important in terms of catalytic applications, and therefore we included in our review the results of XPS studies for each paper, in which such data were available. However, we also found that a few papers provided important pieces of information, even without XPS data, and therefore we also discussed their results in our review. For a better overview and for the sake of clarity, we highlighted the sections containing XPS data in the revised manuscript.
  2. According to the Reviewer’s suggestion, we included the general mechanism of the Cu-catalyzed Ullmann homocoupling reaction in the revised paper, together with an interpretation (see page 2, highlighted).
  3. The Figure caption of Scheme 10, which has been indicated as Scheme 11 in the revised version, has been modified as requested.

We thank the Reviewer again and hope that he/she will find the revised version of our manuscript satisfactory.

Reviewer 2 Report

The Ullmann homocoupling reaction has been found to play an important role in organic synthesis and medicinal chemistry.  In this review, the author summarized the current results on the Ullmann homocoupling reaction with an emphasis on the development of novel catalytic systems: heterogeneous Pd catalysts offer enhanced performances as compared with those of copper-based catalysts; the progress development of gold nanoparticles, bimetallic systems, and ligand-free catalysts in Ullmann homocoupling reaction. Also, a future perspective development has been emphases on improving catalytic systems with novel ligands, anchoring species, and/or support materials, and catalyst recycling. Therefore, this reviewer would like to recommend this manuscript be published on molecules.

Author Response

We would like to thank the Reviewer for appreciating our work and we understand that he required no modifications in the manuscript.

Reviewer 3 Report

1.     “Sarkar et al. developed a novel synthesis procedure” what’s the “novel”. It is only new method.

2.     The syntheses on heterogeneous catalytic systems should be cited, such as Polym. Chem., 2022, 13, 2351–2361; Chem. Commun., 2022, 58, 6653–6656; Org. Chem. Front., 2020,7, 3515-3520; New J. Chem., 2020, 44, 16265-16268; J. Org. Chem. 2019, 84, 14627−14635 and Org. Chem. Front., 2021, 8, 4554–4559

3.     What potential does further research hold? What is the ultimate goal in this field?

4.     Does the future of study lie in this area? Are there other more promising areas in the field which could be progressed?

5.     How will the field evolve in the future? In your perspective, what will the standard procedure have gained or lost from the current norm in five or ten years?

6.     The keywords should be simplified.

7.     Please also list a Table for comparing the previous on this work and heterogeneous catalytic systems.

Author Response

We would like to thank the Reviewer for evaluating our manuscript. The answers to his questions and comments are as follows.

  1. In the sentence …”Sarkar et al. developed a novel method”, the word ”novel” has been replaced by ”new” (see page 2, highlighted).
  2. We appreciate the Reviewer’s suggestions concerning additional References; however, we feel that the suggested References do not fit into the scope of our manuscript, which is focused on the Ullmann homocoupling reaction. All papers listed by the Reviewer discuss other reactions, g. the conjugated addition of para-quinone methides with β-ketoesters (Chem. Commun. 58, 6653-6656, 2022), the sodium triethyl borohydride-catalyzed reduction of unactivated amides to secondary or tertiary amines (J.Org. Chem. 84, 14627-14635, 2019), or the hydroboration of amides to amines (Org. Chem. Front. 7, 3515-3520).
  3. The questions listed in sections 3, 4 and 5 are very similar. The future perspective of the research area discussed in our manuscript has already been mentioned in the Conclusions. Nevertheless, according to the Reviewer’s suggestion, we have completed this issue by adding some more aspects in the revised paper (see the Conclusions, page 20, highlighted).
  4. As suggested by the Reviewer, the Keywords have been simplified in the revised paper (see page 1, highlighted).
  5. It is not clear what the Reviewer meant in section 7. We think that in a mini review, focused on the Ullmann homocoupling reaction, there is no place for a comparative Table listing previous results obtained for heterogeneous catalytic systems, especially that the latter systems have not been specified and may include hundreds or more examples. According to our literature survey, the addition of such Tables has been more common and more appropriate in individual papers, discussing results obtained for catalysts synthesized by novel methods. Namely, in such cases, the catalytic performance of the new catalyst can be compared with those obtained for previously applied and/or conventional catalytic systems (see, g. Table 5 in Ref. 43 in the revised manuscript). However, this is not the subject of our current mini review, and therefore we would prefer not to include such Table in the revised version.

Finally, we thank the Reviewer again for his comments and hope that he will find the modifications in our revised paper satisfactory.

Round 2

Reviewer 3 Report

ok